# Managing Financial Performance toward Achievements in Sustainability Prospects: Comparative Analysis of the e-Commerce and Hospitality Industries

Gabrijela Velickovic, Jelena Stanojevic * and Milan Veselinovic *

International Business College Mitrovica, 40000 Mitrovica, Kosovo; g.velickovic@ibcmitrovica.eu
* Correspondence: j.stanojevic@ibcmitrovica.eu (J.S.); m.veselinovic@ibcmitrovica.eu (M.V.)

**Abstract:** The recent pandemic has been identified as a driver of one of the most severe socio-economic crises over the last few decades. While some sectors have experienced an expansion, others have struggled with a changed business environment. The aim of this research is to simultaneously examine the financial performance and sustainability of the e-commerce and hospitality industries, applying asset and debt ratio analysis to the top five companies in the world from each sector in the time period from 2017 to 2022. The results indicate that the assessed companies demonstrated the ability to successfully manage some of their assets. The debt ratio analysis implies that the assessed companies in the hotel industry have reshaped their capital structure, increasing their reliance on debt in 2020 and 2021 to finance their assets. On the contrary, the selected e-commerce companies were found on average to rely less on debt to finance assets. In accordance with expectations, the differences across the examined sectors and companies that have been observed are mostly in regard to the lower scale of utilization of fixed assets to generate turnover, and in terms of the increased share of debts used to finance assets in the hotel industry, which was among the first and hardest hit by the pandemic. Consequently, the study allows policy makers to identify distinctive strategies for each area of economic activity.

**Keywords:** e-commerce; hospitality; COVID-19; asset management; debt management; financial performance

## 1. Introduction

The COVID-19 pandemic has been qualified as the event that induced one of the most severe socio-economic crises over the last decades. As such, apart from the health cataclysm, the coronavirus outbreak also triggered voluminous distortions to the world economy, creating enormous uncertainty with long-term impacts on all business operations across global industries (Panopio and Cudia 2022). The economic turmoil, triggered by social distancing and other government-imposed measures to fight COVID-19, was stimulated by a negative shock reflected in both the demand and supply sides. Strong movement restrictions, together with business closures, have led to a meaningful decline in the availability and sales of goods and services. Reinforcing each other, these shocks have brought many challenges to the functioning of supply chains, international trade, investment flows and labor markets worldwide. Such adversity, embodied in resource deficiency, steep drops in the volume of operations, collapses in consumption, and hence plunges in sales turnovers and cash flows alike, has had serious repercussions on corporate finance and financial sustainability (Baldwin and Weder di Mauro 2020; Gretzel et al. 2020; Kowalewski and Śpiewanowski 2020). Under these circumstances, making the right financial management decisions toward securing sustainable finance should be at the heart of businesses' operational strategies and their attempted alterations. Setting the financial sustainability objectives as a priority and enforcing sustainable maintenance practices should help businesses in controlling their investments, expenses and income, thus directly

contributing to their cost reduction and risk mitigation, and maximizing the lifecycle of their assets and the value extraction from tangible assets. The accomplishment of financial safety goals hence not only ensures their market presence in turbulent times, but also creates the potential for strategic growth in the long run (Nobanee 2021).

Even though the International Monetary Fund, assessing declines in economic growth (−3.2%) and international trade (−8.1%) in 2020, has disclosed optimistic projections for the aftermath of the crisis, the recent pandemic has still left a deep scar in some industries, thus making their recovery subject to numerous discussions (International Monetary Fund 2021). Despite the undisputable negative effects of this crisis in all sectors, some industries were clearly more challenged than the others, which leads one to a conclusion that the impact of the pandemic was asymmetric. Accordingly, with the imposed travel limitations and lockdowns, the airline and tourism industry, being challenged to adhere to social distancing measures more rigorously than other sectors, experienced the most severe plunge, experiencing an enormous cut in their operations, as well as job losses and a decline in sales revenues (Im et al. 2021; Pagano et al. 2020). In this regard, as assessed by the World Tourism Organization (2020), there was recorded a 44% decrease in international tourist arrivals in 2020 Q1 compared with the same period in 2019, which resulted in a USD 195 billion loss of revenues. Such devastating effects on the tourism industry worldwide, however, showed some differences between geographical regions. Asia and the Pacific, which were the first to be affected by COVID-19, experienced a 51% drop in arrivals in 2020 Q1, followed by Europe, with a 44% decrease in arrivals, the Middle East (−40%), the Americas (−36%) and Africa (−35%) (World Tourism Organization 2020).

On the other hand, online retailers and software providers have experienced a significant rise in their service provision, thus benefiting from the changing economic circumstances. Having the intention of fitting into new business frameworks has resulted in the flourishing of e-business models that have become an inevitable mode of "doing business", even among traditional retailers. With the significant limitations of conventional trade transactions imposed by anti-pandemic measures, e-commerce has become a reasonable alternative in the light of the recent pandemic (Zou and Cheshmehzangi 2022). This fascinating boom in e-commerce, triggered by the new technological revolution and accelerated by the COVID-19 outbreak, could be explained in relation to the benefits, reflected in increased cost efficiency, better information management and supplier integration, and market redefining and growth, which all together bring voluminous opportunities for enhanced turnover and better financial performance in the strongly competitive global market (Xia and Zhang 2010; Cosgun and Dogerlioglu 2012; Damanpour and Damanpour 2001). Remarkable results and the expansion of the e-retail business could be extensively attributed to a rise of e-commerce platforms such as Amazon, Alibaba, MercadoLibre, Jumia, Walmart and others (World Bank 2020).

Nevertheless, despite the many opportunities that arise from the adoption of e-business models, not all market players could benefit from this. While an important rise in this regard has been recorded in China and a few other developed markets, mostly in the fields of the food supply chain, ICT equipment, personal protection and home activities, a lack of the required infrastructure, digital skills, consumer protection and trust have limited such a scenario and the potential of e-commerce to safeguard the economy during the pandemic in developing countries (OECD 2020). Accordingly, one may anticipate that the fertility of e-business models adoption differs across sectors and countries, pointing out that e-retailers may also have been challenged with the same difficulties as traditional sellers during the pandemic, resulting in a decline in their revenues due to an immense cut in overall spending.

Having said the above, it is reasonable to anticipate that the severity of the damage across industries has been driven by their vulnerability and capacity to cope with the new business environment. Given these asymmetries, reconsidering their business models and strategies has emerged as a priority for many players aiming to meet customer needs and secure their existence in light of the global market reshape caused by COVID-19 (Barrero

et al. 2020; Bloom et al. 2020; Deloitte 2020; Denger 2020). Accordingly, the impact of the recent pandemic on particular sectors still remains a black box, despite a variety of expectations. Nevertheless, the literature is scarce in terms of reports that simultaneously examine the impacts of the recent pandemic on the financial performance and sustainability of those sectors, such as e-commerce and hospitality, who have an asymmetric capacity to safeguard their finances.

Aiming to fill this gap, this study intends to assess the effects of COVID-19 on the financial performance and sustainability of the top five companies in the world in the two respective sectors. This study focuses on financial sustainability, being a part of the three-pillar sustainability concept described in the Brundtland Commission Report (1987), which aimed at ensuring the long-term financial security of the observed companies and sustainable value creation (United Nations 1987). Specifically, the authors analyzed the impacts of the pandemic on the key financial indicators from the point of view of asset and debt management. This research focused on the top five companies from both industries, in the time period from 2017 to 2022, intending to examine to what extent the observed companies use financial leverage to finance assets, whether they hold a sufficient amount of different types of assets, as well as how effectively they manage assets in order to generate sales. The findings of this study are intended to provide an in-depth understanding of the two sectors' financial vulnerability and their responses to the reshaped business environment and associated risks, which may be highly relevant to the observed companies' approach to management as they attempt to ensure positive financial results, as well as being relevant to risk-averse investors making investment decisions, and to policy-makers, aiming to mitigate new risks and assess the future prospects of the companies, entire sectors and economies. By doing so, this paper aims to provide valuable insights into the economic consequences of the COVID-19 pandemic on the hotel and e-commerce industries, and to identify potential strategies for recovery and resilience in the face of future crises.

The remaining part of the paper is structured as follows. The following section provides a theoretical background and review of the relevant literature on the impact of COVID-19 on the hospitality and e-commerce industries. The third section of this manuscript briefly describes the research questions and the methodology used to address them. The fourth section provides insights into the main results of this study, while the final section provides the concluding remarks.

## 2. Literature Review

The COVID-19 pandemic and the imposition of related government measures brought about many challenges to all businesses, who faced a sharp decline in purchases and, consequently, who have reasonable concerns about their sales, turnover and profitability (Crets 2020). Along with a decrease in operations, a cut in the firms' operational costs has been anticipated. However, such a scenario turned out to be unlikely, which could be attributed to low cost-to-revenue elasticity, as a significant share of the fixed costs (such as wages and charges) could not be significantly decreased in the short run. Such "sticky" cost behavior has further contributed to the vanishing profitability margins (Golden et al. 2020; Cannon 2014; Shust and Weiss 2014). Together with the plunge in revenues, there was a foreseen short fall in business cash inflows, which, by damaging the cash reserves intended for covering emerging losses, is expected to more profoundly challenge the firms' liquidity, thus raising new obstacles in meeting the current liabilities, such as salaries, and supplier and tax payables. With the intention to manage such increased liquidity needs and meet arising financial obligations, the companies could place themselves in further debt, thus causing their future financial position to become questionable (Putri and Rahyuda 2020; D'Amato et al. 2020; Padoveze and Benedicto 2014; Ellis 2021; Mirza et al. 2020; Shoukry 2020; Devi et al. 2020; Tavares et al. 2023). Given this scenario, maintaining financial sustainability has become one of the top goals of companies across all industries. Worldwide, financial sustainability is considered to be a widely welcomed concept in both companies and society as a whole. As such, it has been given significant consideration by scholars, particularly in

uncertain times (Gregory et al. 2013; Gómez-Bezares et al. 2017; Henock 2019; Zabolotnyy and Wasilewski 2019). According to the concept defined in the Brundtland Commission of the United Nations report (1987), financial management should ensure present financial success without harming the potential for financial accomplishments in the future (United Nations 1987). In line with that, companies striving for strategic growth have identified financial sustainability as a prioritized objective—equally important in turbulent times, as it leads to reducing solvency and refinancing risks—along with embracing risk-adjusted excess returns (Gleißner et al. 2022).

The recent pandemic had the most dramatic consequences for tourism and affiliated sectors, which, being among the first to be hit by the crisis, suffered the most in adhering the social distancing measures (Im et al. 2021; Pagano et al. 2020; Higgins-Desbiolles 2020; McLaughlin 2020). Being faced with a sharp decline in demand, which is especially sensitive to safety and health risks, the sector recorded an immense fall in its activities and job losses, thus bringing many businesses to the edge of their existence (Maniga 2020; Blake et al. 2003; Kozak et al. 2007; Cartwright 2000). As shown by WTO statistics, the tourism sector experienced a decline of 20 million tourists in 2020 Q1 across the globe. The most severe impact was borne by Asia (−64%), followed by Europe (−60%), America (−46%), Africa (−44%) and the Middle East (−41%), which in turn implied a sharp decline in travel spending (−42%) and revenues (-USD 300–450 billion), as well as financial losses, estimated at USD 1.9 trillion in the top 10 tourism markets. As a consequence, 8.3 million staff releases have been recorded, as a result of which both businesses and employees suffered from a significant cut in their income (World Tourism Organization 2020). In accordance with the above, one may anticipate that the recent pandemic and resulting circumstances have given rise to a number of obstacles for the companies' financial managers, who have been challenged to identify the opportunities and define strategies that would repair their future finance prospects, toward revenue increases, liquidity strengthening and optimum asset utilization in the damaged business environment.

Nevertheless, given the fact that with COVID-19, e-purchase has become the only viable model for meeting customers' consumption needs, the new business circumstances have become a fertile ground for the online retail industry, which recorded an extraordinary acceleration in its operations. In line with that, e-commerce has experienced growth of 174%, driven by China as the greatest market, followed by the US, Japan, the UK and Germany. Accordingly, the top nine e-retail companies recorded a double-digit growth in their sales revenues, led by Amazon, which saw a 40% increase in their net sales during the pandemic, which supports the bright expectations for the e-business sector in the COVID era, in relation to its earlier expansion (Statista 2020; Global Data 2020). Launching or prioritizing e-commerce as a dominant business model has therefore enabled many businesses to maintain their operations under new circumstances, allowing the customers access to a wider offer while having their need for safety satisfied. Moreover, the boom in the e-retail industry has given rise to new firms, products and customer segments, thus enhancing the scope and dynamism of the e-commerce sector. Accordingly, the promising trends in the e-business industry during the pandemic have opened the door for many businesses toward an expansion of their operations and sales, thus enabling their impressive financial performance (World Tourism Organization 2020). Still, as this expansion in e-retail has varied across sectors and countries, being particularly limited in developing countries, there has been no common stance in regard to the financial performances of the e-businesses assessed in the light of COVID-19.

Given its particular sensitivity to pandemic conditions, the impact of COVID-19 on the tourism sector has become a topic of particular interest to many researchers aiming to assess the effects of induced crises on a variety of aspects and subsectors in this broad industry. In line with assumptions, a number of empirical studies have confirmed the devastating impacts of the recent pandemic on the hospitality sector, pointing out that the survival of the firms will strongly depend on their financial strength (Clark et al. 2021; Crespí-Cladera et al. 2021; Wieczorek-Kosmala 2021). In relation to this, Foo et al. (2021),

assessing the effects of COVID-19 on the Malaysian market, confirmed the substantial damage undertaken in this sector being a result of a remarkable decline in the number of tourist visits. Accordingly, Bouarar et al. (2020) suggested that the sharpest decline in the revenues was experienced by countries whose economic growth is predominantly driven by the tourism sector. Thus, while the most harmful consequences have been identified in some Pacific Ocean and Asia countries, such as China, no notable effects have been noticed in Algeria, which generates no sizable revenues from tourism. The tourism sectors of the most affected countries have suffered not only due to a drastic decline in the revenue side of their profitability function, but also due to a significant rise in their costs, being induced by complying with required health and safety protocols, enhanced cleaning and sanitary activities, and other personal protection needs. Similarly, some authors have pointed out the diversity of vulnerability across different accommodation types during the pandemic. Accordingly, while hotels have been affected more notably by the crisis, flat rentals have recorded more positive trends (Bresciani et al. 2021; Dolnicar and Zare 2020). The torment experienced by the hospitality sector during the COVID-19 era was also the focus of other authors, who assessed its negative repercussions for countries' regional ecosystem and quality of life (Đorđević et al. 2022; Matteucci et al. 2021), the tourism firms' stock returns (Jordà et al. 2020; Liu et al. 2020; Al-Awadhi et al. 2020), the process of hotel supply chain management (Aigbedo 2021; Milovanović et al. 2021), the hotels' return measures in the aftermath of the pandemic, and the need for upgrading their crisis management processes (Hao et al. 2020; Permatasari and Mahyuni 2022). In this light, undertaking an assessment of the recuperation strategies applied by the businesses in this sector, Özatay and Sak (2020) pointed out that rather than cutting the number of their employees, the firms should consider other strategies for maintaining their existence. Struggling to survive in such difficult circumstances, the most severely affected should be supported by government measures related to tax reduction, employee retention programs, easier access to additional finance, postponing loan servicing, and other financial impulses.

In an attempt to revive their operations and boost revenues, businesses in the tourism industry are advised to place emphasis on a more profound understanding of different customer segments' changing needs, travel behavior, perceived values and greater communication quality, rather than on discounts, which are proven to not be effective under the new circumstances (Singh 2020). Furthermore, implementing new business models and strategies grounded on technological innovations, decreased risks and rebuilding customer perceptions of safety is seen as a priority for companies in the tourism industry in the light of the pandemic (Shin and Kang 2020). Following these recommendations, some hotels have focused on reinforcing local and regional weekend offers, while others went with reshaping their facilities toward "home offices" for traveling workers. In addition, many hotels introduced greater hygiene conditions, more flexibility and a range of innovative practices such as free cancelations or changes of accommodation dates, contactless payment transactions and customer self-service supported through apps for food ordering and laundry services. Still, despite these measures, maintaining operations under the conditions of the plunge in revenues and increased costs during the pandemic has become an unachievable goal for many, particularly for small hotels, which were brought to the edge of their existence and were forced to close their facilities (Singh 2020).

Differently from the tourism sector, the e-business industry was expected to flourish following the COVID-19 outbreak. Several studies investigating the effects of the pandemic on e-commerce have found evidence in support of this expectation. Accordingly, Alfonso et al. (2021), who conducted a comparative analysis of COVID-19's effects on selected countries, pointed out that the rise in e-commerce transactions was found to be stronger in countries with more rigid social distancing measures and in those where e-commerce is an emerging business approach. In this view, Boldea and Boldea (2021), who assessed the relationship between the pandemic's effects and e-market behavior in Romania, employed nonlinear regression and correlation analyses, providing evidence of the significant positive association between the observed variables and confirming a rise in e-retail sales volume.

Following a similar approach, Mandasari and Pratama (2020) identified a positive and strong impact of the pandemic on the sales and income of MSMEs, thus confirming that COVID-19 contributed remarkably to the e-business boom. Since such identified changes in consumer behavior and preferences related to the pandemic are assessed to be long-lasting, e-commerce is seen as a consequential substitute for traditional retail (Bhatti et al. 2020).

Nevertheless, despite the promising expectations for the e-business industry in light of COVID-19, there are findings that reveal that the reality is much more compound (Sterling 2020; Al Hamli and Sobaih 2023). This view is supported by the fact that an expansion of the demand for e-commerce transactions may result in the extension of their completion time, introducing risks to distribution, poor customer experiences, and potential cuts to future demand. Moreover, the challenges for e-business could become even worse, given that travel limitations, shipping issues and closures of factories in China could result in inventory and product shortages, which, together with decreased customer demand, will create obstacles for e-business growth (Sterling 2020; Crets 2020; Porter 2020). Accordingly, e-retailers such as Amazon, FedEx and UPS have experienced postponements of order completion, which, as noted by Porter (2020), could be explained by the enormous increase in demand and shipping delays during the pandemic. In addition to the fact that COVID-19 abruptly contributed to the vanishing of profit margins in traditional retail, such a negative relationship was also revealed in the study performed by Alvarez and Marsal (2021), who identified a negative association between the share of e-sales and profit margins for five European markets (France, Germany, Italy, Switzerland and the UK).

Considering the above findings, one could conclude that COVID-19 had an asymmetric impact on the two observed industries, giving rise to the opposing beliefs in terms of their prospects in the light of the pandemic. While there is no doubt that tourism and its affiliated sectors have been among the hardest hit industries worldwide, challenging them to find the modality for preserving their operations and finances, such circumstances have unlocked a number of opportunities for the reshaping of their businesses and innovations directed toward enhancing customer experience and reviving their incentives to travel. However, despite the implementation of these measures, this sector and its finance recuperation model are influenced by many factors, among which is the need to rebuild a feeling of safety.

On the other hand, given the comfort and numerous advantages related to e-purchase, one may conclude that e-commerce expansion during COVID-19 has permanently reshaped consumer purchase behavior, thus giving a rise to its irreversible growth and the financial success of e-businesses in the long run, beyond the pandemic (OECD 2020). Still, the effects of COVID-19 on e-commerce and its finance remain a black box, which could be attributed to the fact that launching an innovative and auspicious business model is not sufficient for guaranteed success, particularly during a crisis. Such a scenario rather requires the composite interaction and suitability of various resources and business capacities (Andonov et al. 2021). With the aim of mitigating negative repercussions and benefiting from circumstances arising in the time of the pandemic, e-businesses need to ensure high flexibility, low prices and fast shipping, as well as quality customer service and high satisfaction with customer experience.

## 3. Methodology and Research Results

The aim of the research is to simultaneously examine the financial performance and sustainability trends of the e-commerce and hospitality industries in the last few challenging years, this being the period affected by COVID-19 as well as fluctuations in the economic market due to socio-political changes. While some industries have experienced an expansion, other industries have struggled with a changed business environment. Thus, the aim of the comparative analysis of e-commerce, as a promising industry, and hospitality, as an industry damaged by the crisis, is to explore the similarities and differences in managing financial performance from the points of view of asset and debt management within the area of business sustainability. Asset management analysis has been conducted in order to evaluate the risks related to companies' assets, and to determine which assets

are precious and should be preserved, but also which assets face potential risks and deserve additional attention. Debt management analysis, in addition, provides a better understanding of the financial leverage magnitude and possible impacts of companies' debt on their financial sustainability. Both assets and debt are the most broadly researched and most relevant ratios, with their analysis serving to provide an insight into the companies' ability to effectively use and finance their assets in challenging periods.

Research has been conducted on the top five companies in the world from both industries, in the time period from 2017 to 2022, intending to examine to what extent companies use financial leverage to finance assets, whether they hold a sufficient amount of different types of assets, as well as how effectively they manage assets in order to generate sales. Based on the (World Bank 2023) database, the top five companies from the hospitality industry, with the highest value of turnover in 2022, are the following: Marriott International Inc. (20.8 billion), Hilton Worldwide Holdings Inc. (8.8 billion), Hyatt Hotels Corporation (5.9 billion), Host Hotels & Resorts, Inc. (4.9 billion) and InterContinental Hotels Group (3.9 billion). According to the same criteria and source, the top five e-commerce companies are Amazon (513.9 billion), Alphabet (282.8 billion), JD (151.7 billion), Alibaba (134.6 billion) and Meta (116.6 billion).

The analysis of financial performance and sustainability, via asset and debt management, uses financial statements of the selected companies as the information base. Income statements and balance sheets, available for external stakeholders on the companies' websites, provide all relevant information for the research. Following the main analysis of asset and debt management for each of the selected companies within the hospitality and e-commerce industries, their comparative analysis is of crucial importance to our main research goal, and provides input for the following research questions:

1. Aiming to ensure financial sustainability, do the selected hospitality and e-commerce companies effectively use their tangible and intangible assets to generate sales in the whole observed period?
2. Did the companies, from both e-commerce and hospitality industries, offer receivable privileges to their customers in the time of the economic crisis in 2020?
3. Have the e-commerce and hospitality companies managed to sustain the desired capital structure, changing their financial leverage in the same direction over the analyzed period, especially in the last three years?

To ensure clear organization and discussion, the research results have been separated into two parts: asset and debt management.

Asset management analysis enables managers and investors to assess how efficiently companies use assets (inventory, accounts receivable, fixed asset) (Cornett et al. 2022). The most commonly used asset management ratios, being applied in this research, are grouped by type of asset:

4. Inventory management:
    - Inventory turnover ratio = Net Sales or Revenue/Inventory
    - Days' sales in inventory = Inventory ∗ 365 days/Net Sales or Revenue
5. Accounts receivable management:
    - Average collection period = Accounts Receivable 8 365 days/Net Sales or Revenue
    - Accounts receivable turnover = Net Sales or Revenue/Accounts receivable
6. Fixed asset and working capital management:
    - Fixed asset turnover ratio = Net Sales or Revenue/Fixed Assets
    - Sales to working capital = Net Sales or Revenue/Working Capital
7. Total asset management:
    - Total assets turnover ratio = Net Sales or Revenue/Total Assets
    - Capital intensity ratio = Total Assets/Net Sales or Revenue

The asset management ratios of the examined hospitality and e-commerce companies are shown in Table 1.

**Table 1.** Asset management ratios for the e-commerce and hospitality companies, 2017–2022.

| | Companies | Asset Management Ratios | 2022 | 2021 | 2020 | 2019 | 2018 | 2017 |
|---|---|---|---|---|---|---|---|---|
| E-commerce | Amazon | | 14.94 | 14.39 | 16.22 | 13.69 | 13.56 | 11.08 |
| | Alphabet | | 105.93 | 220.20 | 250.72 | 162.02 | 123.59 | 148.00 |
| | JD | | 12.79 | 16.53 | 13.74 | 12.94 | 10.48 | 13.37 |
| | Alibaba | | n/a | n/a | n/a | n/a | n/a | n/a |
| | Meta | Inventory turnover ratio | n/a | n/a | n/a | n/a | n/a | n/a |
| Hotels | Marriott | | n/a | 55.21 | 58.73 | 41.36 | 80.77 | 53.26 |
| | Hilton | | n/a | 28.65 | 43.95 | 131.28 | 52.70 | 48.11 |
| | Hyatt | | 654.56 | 302.80 | 229.56 | 418.33 | 318.14 | 318.71 |
| | Host Hotels & Resorts | | n/a | n/a | n/a | n/a | n/a | n/a |
| | InterContinental | | 973.00 | 969.00 | 598.50 | 925.40 | 813.25 | 1582.00 |
| E-commerce | Amazon | | 24 | 25 | 22 | 27 | 27 | 33 |
| | Alphabet | | 3 | 2 | 1 | 2 | 3 | 2 |
| | JD | | 29 | 22 | 27 | 28 | 35 | 27 |
| | Alibaba | | n/a | n/a | n/a | n/a | n/a | n/a |
| | Meta | Days' sales in inventory | n/a | n/a | n/a | n/a | n/a | n/a |
| Hotels | Marriott | | n/a | 7 | 6 | 9 | 5 | 7 |
| | Hilton | | n/a | 13 | 8 | 3 | 7 | 8 |
| | Hyatt | | 1 | 1 | 2 | 1 | 1 | 1 |
| | Host Hotels & Resorts | | n/a | n/a | n/a | n/a | n/a | n/a |
| | InterContinental | | 0.38 | 0.38 | 0.61 | 0.39 | 0.45 | 0.23 |
| E-commerce | Amazon | | 30 | 26 | 23 | 27 | 26 | 27 |
| | Alphabet | | 52 | 56 | 63 | 62 | 57 | 62 |
| | JD | | 4 | 3 | 3 | 7 | 14 | 16 |
| | Alibaba | | n/a | n/a | n/a | n/a | n/a | n/a |
| | Meta | Average collection period | 42 | 43 | 48 | 49 | 50 | 52 |
| Hotels | Marriott | | 45 | 52 | 61 | 42 | 38 | 38 |
| | Hilton | | 55 | 67 | 65 | 49 | 48 | 50 |
| | Hyatt | | 52 | 76 | 56 | 31 | 35 | 35 |
| | Host Hotels & Resorts | | 38 | 14 | 5 | 4 | 5 | 5 |
| | InterContinental | | 46 | 49 | 54 | 35 | 48 | 51 |
| E-commerce | Amazon | | 12.13 | 14.28 | 15.73 | 13.48 | 13.96 | 13.51 |
| | Alphabet | | 7.03 | 6.55 | 5.82 | 5.89 | 6.46 | 5.93 |
| | JD | | 81.25 | 137.00 | 128.54 | 51.28 | 26.73 | 22.14 |
| | Alibaba | | n/a | n/a | n/a | n/a | n/a | n/a |
| | Meta | Accounts receivable turnover | 8.66 | 8.40 | 7.58 | 7.43 | 7.36 | 6.97 |
| Hotels | Marriott | | 8.08 | 6.99 | 5.98 | 8.76 | 9.73 | 9.69 |
| | Hilton | | 6.61 | 5.42 | 5.59 | 7.50 | 7.61 | 7.33 |
| | Hyatt | | 7.06 | 4.78 | 6.54 | 11.92 | 10.43 | 10.30 |
| | Host Hotels & Resorts | | 9.68 | 25.58 | 73.64 | 86.81 | 77.80 | 67.34 |
| | InterContinental | | 7.94 | 7.40 | 6.80 | 10.37 | 7.55 | 7.19 |
| E-commerce | Amazon | | 1.63 | 1.81 | 2.05 | 2.18 | 2.66 | 2.50 |
| | Alphabet | | 1.41 | 1.51 | 1.26 | 1.31 | 1.41 | 1.52 |
| | JD | | 4.91 | 5.20 | 6.60 | 5.46 | 6.33 | 7.24 |
| | Alibaba | | 0.81 | 0.69 | 0.60 | 0.54 | 0.54 | 0.49 |
| | Meta | Fixed asset turnover ratio | 0.92 | 1.19 | 1.03 | 1.05 | 1.19 | 1.13 |
| Hotels | Marriott | | 0.97 | 0.63 | 0.48 | 0.96 | 0.99 | 0.97 |
| | Hilton | | 0.69 | 0.46 | 0.34 | 0.73 | 0.74 | 0.67 |
| | Hyatt | | 0.59 | 0.29 | 0.31 | 0.75 | 0.71 | 0.71 |
| | Host Hotels & Resorts | | 0.45 | 0.26 | 0.16 | 0.52 | 0.54 | 0.54 |
| | InterContinental | | 1.53 | 1.10 | 0.86 | 1.52 | 1.52 | 1.99 |

**Table 1.** *Cont.*

| | Companies | Asset Management Ratios | 2022 | 2021 | 2020 | 2019 | 2018 | 2017 |
|---|---|---|---|---|---|---|---|---|
| E-commerce | Amazon | Sales to working capital | −59.75 | 24.33 | 60.82 | 32.92 | 34.71 | 76.87 |
| | Alphabet | | 2.96 | 2.08 | 1.55 | 1.51 | 1.35 | 1.11 |
| | JD | | 12.39 | 16.03 | −862.57 | −35.59 | −135.70 | 176.41 |
| | Alibaba | | 3.35 | 2.70 | 2.31 | 6.02 | 2.07 | 1.78 |
| | Meta | | 3.59 | 2.59 | 1.42 | 1.38 | 1.28 | 0.91 |
| Hotels | Marriott | | −5.16 | −4.98 | −3.61 | −5.91 | −5.56 | −6.67 |
| | Hilton | | −17.48 | −39.11 | 2.43 | −12.15 | −14.09 | −17.87 |
| | Hyatt | | −5.68 | −17.81 | 1.31 | 8.10 | 15.68 | 13.32 |
| | Host Hotels & Resorts | | −83.17 | 5.44 | 1.83 | 9.77 | 3.05 | 4.30 |
| | InterContinental | | 29.04 | 6.58 | 6.37 | −10.24 | −104.94 | −10.11 |
| E-commerce | Amazon | Total assets turnover ratio | 1.11 | 1.12 | 1.20 | 1.25 | 1.43 | 1.35 |
| | Alphabet | | 0.77 | 0.72 | 0.57 | 0.59 | 0.59 | 0.56 |
| | JD | | 1.95 | 2.31 | 3.06 | 2.72 | 2.38 | 2.41 |
| | Alibaba | | 0.50 | 0.42 | 0.39 | 0.39 | 0.35 | 0.31 |
| | Meta | | 0.63 | 0.71 | 0.54 | 0.53 | 0.57 | 0.48 |
| Hotels | Marriott | | 0.84 | 0.54 | 0.43 | 0.84 | 0.88 | 0.86 |
| | Hilton | | 0.57 | 0.37 | 0.26 | 0.63 | 0.64 | 0.57 |
| | Hyatt | | 0.48 | 0.24 | 0.23 | 0.60 | 0.58 | 0.59 |
| | Host Hotels & Resorts | | 0.40 | 0.23 | 0.13 | 0.44 | 0.46 | 0.46 |
| | InterContinental | | 0.92 | 0.62 | 0.48 | 1.16 | 1.01 | 1.42 |
| E-commerce | Amazon | Capital intensity ratio | 0.90 | 0.90 | 0.83 | 0.80 | 0.70 | 0.74 |
| | Alphabet | | 1.29 | 1.39 | 1.75 | 1.70 | 1.70 | 1.78 |
| | JD | | 0.51 | 0.43 | 0.33 | 0.37 | 0.42 | 0.41 |
| | Alibaba | | 1.99 | 2.36 | 2.58 | 2.58 | 2.87 | 3.20 |
| | Meta | | 1.59 | 1.41 | 1.85 | 1.89 | 1.74 | 2.08 |
| Hotels | Marriott | | 1.19 | 1.84 | 2.34 | 1.19 | 1.14 | 1.17 |
| | Hilton | | 1.77 | 2.67 | 3.89 | 1.58 | 1.57 | 1.75 |
| | Hyatt | | 2.09 | 4.16 | 4.42 | 1.68 | 1.72 | 1.70 |
| | Host Hotels & Resorts | | 2.50 | 4.27 | 7.96 | 2.25 | 2.19 | 2.17 |
| | InterContinental | | 1.08 | 1.62 | 2.10 | 0.86 | 0.99 | 0.70 |

Source: Authors' calculations.

1. Inventory turnover ratio, which belongs to the first group out of the four asset management ratios, indicates that the highest level of sales per dollar of inventory in the hospitality area was achieved by InterContinental (973 in 2022), and among the e-commerce companies was achieved by Alphabet (106 in 2022). Right after InterContinental, according to this ratio, is Hyatt (655 in 2022), and this ratio for both hotel companies (except Hyatt in 2020 and 2021) is several times higher than that for Alphabet. While InterContinental and Hyatt decreased the value of their sales per dollar of inventory in 2020 as a consequence of COVID-19, Alphabet and Amazon, belonging to the e-commerce sector, managed to achieve the highest value of this ratio in 2020. The least number of days for which the inventory is held before the final product is sold was show by Alphabet among the e-commerce companies, and InterContinental and Hyatt in the second group. Therefore, these companies managed to turn inventory into sold products as quickly as possible (Alphabet 1–3 days, Hyatt 1–2 days and InterContinental less than 1 day), reducing warehousing, monitoring, insurance, and all other expenses related to serving the inventory. However, extremely low days' sales in inventory or high inventory turnover ratios can indicate that a company does not hold sufficient inventory for operations, jeopardizing the company's main activity and causing the waste of fixed resources.

2. Accounts receivable management shows that JD our of the e-commerce companies and Host Hotels & Resorts our of the hospitality companies recorded the highest levels of sales per dollar of accounts receivable, and collected their accounts receivable for 3–4 days

(JD in 2020 and 2021, Host Hotels & Resorts in 2019). Even though it is desirable that the accounts receivable turnover ratio be high and the average collection period (ACP) be low, extreme values of these two ratios can be a sign of bad company management, considering that an overly strict accounts receivable policy can discourage customers and force them to do business with competitors with better credit terms. Of the selected, the only companies that extended their average collection period in 2020 were Alphabet, Marriott and InterContinental, while JD and Amazon reduced the number of days for which accounts receivables are held before they collect cash from the sale.

3. Fixed asset turnover ratio and sales to working capital, which belong to the third group in the asset management analysis, were at their highest values for JD (2017) and InterContinental (2017 and 2021). A higher level of sales per dollar of fixed assets or working capital is a sign of good management. Extreme values of these two ratios, which is not the case in the selected companies, may indicate that the company is reaching or is close to maximum production capacity. Regarding fixed asset turnover ratio, the lowest value among e-commerce companies was reached in 2020 by Alphabet only, while this was the case for another company in 2017 (Alibaba), and the rest of companies in 2022 (Amazon, JD and Meta). As for the hotels, except Hyatt in 2021, all other hotels analyzed recorded the lowest value of this ratio in 2020.

4. The highest number of dollars of sales produced per dollar of total assets, or total assets turnover ratio, was achieved by JD in 2020 (3.06). All other companies, both e-commerce and hotels, recorded the highest values of this ratio (between 0.46 and 1.43) in different years. The least dollars of assets per dollar of sales, or capital intensity ratio, was achieved by JD in 2020 (0.33).

Debt management analysis is used to assess a company's financial leverage (Spiceland et al. 2014), and thereby evaluate to what extent the company relies on debt in its capital structure (Gianfelici and Subramanyam 2019). The debt management ratios, which measure the degree to which companies use debt to finance assets (Penman 2012) and are used in the research, are the following:

- Debt ratio = Total debt/total assets $* 100$
- Debt-to-equity ratio = Total debt/total equity
- Equity multiplier ratio = Total assets/total equity
- Times interest earned = Earnings before interest and taxes (EBIT)/Interest

The debt management ratios used in the research (Table 2) evaluate to what extent the companies finance their assets with debt versus equity, as well as whether the companies generate sufficient income to pay their debts (Cornett et al. 2022). The biggest percentage of total assets financed with debt, among hotels, was shown by InterContinental in 2022, while Marriott, Hilton, Host Hotels & Resorts displayed the maximum values of debt ratio in 2020, and Hyatt in 2021. These hotels increased their reliance on debt in 2020 and 2021 to finance their assets. On the contrary, the selected e-commerce companies relied less on debt on average to finance their assets compared to hotels, and the highest values of this ratio, aside from that achieved by Alphabet in 2020 and Meta in 2022, were reached in 2017 and 2018. The debt-to-equity ratio shows that Marriott had USD 56.44 of debt for each dollar of equity in 2020, which is the highest value of this ratio amongst all selected companies. In the same year, the highest value of this ratio was achieved by Alphabet (0.44). Amazon managed to lower their share of debt in the capital structure from 2017 (3.74) to 2022 (2.17). In general, e-commerce companies have lower debts versus equity in financing assets (only Amazon and JD have a debt-to-equity ratio close to 4). In addition to Marriott, among hotels, Hilton also has a high debt-to-equity ratio, with a 24.08-fold higher debt than equity. Slightly higher debt than equity in the whole analyzed period was recorded for Alphabet, Alibaba, Meta and Host Hotels & Resorts. Given that the total liabilities exceeded total assets, InterContinental in all years had negative equity and, accordingly, debt-to-equity ratio. The least number of dollars per dollar of equity, or equity multiplier ratio, was shown by Meta in 2017 (1.14) out of the e-commerce companies, and by Host Hotels & Resorts (1.61) in 2018 out of the hotels, while the maximum value of this ratio was

recorded by Amazon in 2017 (4.74) and Marriott in 2020 (57.44). Lower values of the three previously explained debt management ratios indicate less debt and more equity used by the companies to finance assets. On the other hand, times interest earned tended towards its highest possible value. That is, in terms of the number of dollars of operating earnings per dollar of interest expenses, Meta recorded the highest value of this ratio in 2022 (231.55) among the e-commerce companies, and Marriott did so in 2022 (9.21) for the hotels.

**Table 2.** Debt management ratios for the e-commerce and hospitality companies, 2017–2022.

| | Companies | Debt Management Ratios | 2022 | 2021 | 2020 | 2019 | 2018 | 2017 |
|---|---|---|---|---|---|---|---|---|
| e-commerce | Amazon | Debt ratio | 68.44 | 67.13 | 70.92 | 72.45 | 73.23 | 78.90 |
| | Alphabet | | 29.87 | 29.96 | 30.37 | 26.99 | 23.70 | 22.70 |
| | JD | | 50.54 | 51.58 | 67.40 | 70.90 | 71.54 | 78.70 |
| | Alibaba | | 36.74 | 36.40 | 33.70 | 36.68 | 39.14 | 36.64 |
| | Meta | | 32.31 | 24.77 | 19.47 | 24.23 | 13.57 | 12.04 |
| Hotels | Marriott | | 97.71 | 94.47 | 98.26 | 97.19 | 90.61 | 84.56 |
| | Hilton | | 107.08 | 105.30 | 108.87 | 103.16 | 96.01 | 88.11 |
| | Hyatt | | 69.93 | 71.71 | 64.79 | 52.87 | 51.89 | 49.12 |
| | Host Hotels & Resorts | | 43.93 | 46.79 | 50.09 | 39.32 | 31.74 | 33.87 |
| | InterContinental | | 138.14 | 131.26 | 136.69 | 136.85 | 127.64 | 143.24 |
| e-commerce | Amazon | Debt-to-equity ratio | 2.17 | 2.04 | 2.44 | 2.63 | 2.73 | 3.74 |
| | Alphabet | | 0.43 | 0.43 | 0.44 | 0.37 | 0.31 | 0.29 |
| | JD | | 1.02 | 1.07 | 2.07 | 2.44 | 2.51 | 3.69 |
| | Alibaba | | 0.58 | 0.57 | 0.51 | 0.59 | 0.64 | 0.58 |
| | Meta | | 0.48 | 0.33 | 0.24 | 0.32 | 0.16 | 0.14 |
| Hotels | Marriott | | 42.69 | 17.07 | 56.44 | 34.63 | 9.65 | 5.63 |
| | Hilton | | −15.13 | −19.85 | −12.28 | −32.69 | 24.08 | 7.41 |
| | Hyatt | | 2.33 | 2.53 | 1.84 | 1.12 | 1.08 | 0.97 |
| | Host Hotels & Resorts | | 0.78 | 0.88 | 1.00 | 0.65 | 0.51 | 0.57 |
| | InterContinental | | −3.62 | −4.20 | −3.73 | −3.71 | −4.59 | −3.30 |
| e-commerce | Amazon | Equity multiplier ratio | 3.17 | 3.04 | 3.44 | 3.63 | 3.73 | 4.74 |
| | Alphabet | | 1.43 | 1.43 | 1.44 | 1.37 | 1.31 | 1.29 |
| | JD | | 2.02 | 2.07 | 3.07 | 3.44 | 3.51 | 4.69 |
| | Alibaba | | 1.58 | 1.57 | 1.51 | 1.60 | 1.64 | 1.58 |
| | Meta | | 1.48 | 1.33 | 1.24 | 1.32 | 1.16 | 1.14 |
| Hotels | Marriott | | 43.69 | 18.07 | 57.44 | 35.63 | 10.65 | 6.66 |
| | Hilton | | −14.13 | −18.85 | −11.28 | −31.69 | 25.08 | 8.41 |
| | Hyatt | | 3.33 | 3.53 | 2.84 | 2.12 | 2.08 | 1.97 |
| | Host Hotels & Resorts | | 1.78 | 1.88 | 2.00 | 1.65 | 1.61 | 1.68 |
| | InterContinental | | −2.62 | −3.20 | −2.73 | −2.71 | −3.59 | −2.30 |
| e-commerce | Amazon | Times interest earned | 0.67 | 1.87 | 17.90 | 25.74 | 10.71 | 13.69 |
| | Alphabet | | 21.30 | 6.55 | 6.01 | 6.35 | 3.72 | 25.79 |
| | JD | | 3.37 | 0.62 | 0.32 | 1.91 | 10.67 | 0.87 |
| | Alibaba | | 2.69 | 1.18 | 1.23 | 1.46 | 2.28 | 4.01 |
| | Meta | | 231.55 | 88.05 | 64.19 | 29.04 | 55.61 | 51.67 |
| Hotels | Marriott | | 9.21 | 4.48 | 0.84 | 5.27 | 7.93 | / |
| | Hilton | | 5.05 | 2.56 | 0.28 | 3.81 | 3.86 | / |
| | Hyatt | | 3.75 | 1.86 | 6.45 | 3.94 | 6.92 | / |
| | Host Hotels & Resorts | | 6.02 | 1.37 | 5.15 | 4.32 | / | / |
| | InterContinental | | 7.16 | 3.64 | 1.25 | 6.64 | / | / |

Source: Authors' calculations.

The six-year analysis of assets and debt management that we undertook, applied to the top five companies in the world from the e-commerce and hospitality industries, responds to the primary research aim and related research questions.

The asset management analysis confirmed the companies' ability to effectively manage some of their assets in the analyzed period from 2017 to 2022. According to the available data, the selected companies achieved sales greater than one per each dollar of inventory (inventory turnover ratio) and accounts receivable (accounts receivable turnover ratio), showing their effective and sustainable use of inventory and accounts receivable. The greatest contribution in generating sales was shown by inventory (e.g., in 2017 for InterContinental, USD 1582 of sales per each dollar of inventory) and accounts receivable (e.g., in 2021, JD managed to have USD 137 of sales per dollar of accounts receivable). In addition, the companies showed a low number of days that inventory is held before the final product is sold, and even the hotel companies kept this ratio at a lower level. When it comes to fixed assets, working capital and total assets, the analyzed companies struggled in generating sufficient sales per each dollar of these assets. That is, in most cases, the companies could not reach sufficient sales to cover investment in these assets, whereby these issues were more pronounced in hotel companies.

Although companies seek to offer better purchase conditions to customers in times of crises, aiming to encourage them to keep buying their goods and services, not all the selected companies offered accounts receivable privileges to their customers in times of crises in 2020. Namely, Alphabet, Marriott and InterContinental were the only ones to do so, with the longest average collection period being in 2020. On the contrary, in 2020, JD and Amazon expected their customers to pay in cash in fewer days than in any other year of the analyzed period. Meta recorded a decreasing trend in the average collection period from 2017 (52 days) to 2022 (42 days), while Hilton, Hyatt and Host Hotels & Resorts generally increased the value of this ratio in the last three years.

The debt management analysis shows that the analyzed companies changed their capital structures over the observed period, with many fluctuations and frequently variable values. Despite this, the conducted research (based on debt ratio, debt-to-equity and equity multiplier ratio) shows that in 2020, Alphabet and Marriott reached the maximum share of debt versus equity in their financing assets. However, the values of these debt management ratios for Marriott were several times higher than those for Alphabet, implying the difficulties that hotel companies have faced in maintaining a sustainable and desired capital structure over the last few years. Hilton and Host Hotels & Resorts also recorded their maximum values for some of these three ratios in 2020. When it comes to the operating earnings available to meet interest expenses, with the times interest earned being less than 1, Amazon, JD, Marriott and Hilton could not achieve sufficient income to cover interest expenses in the last three years. All other evaluated companies succeeded in keeping their times interest earned over 1 in the whole period. JD is the only analyzed company for which this ratio was below 1 in three out of the six analyzed years.

## 4. Conclusions

With the outbreak of COVID-19, maintaining existence in the market has become one of the priorities of businesses in all industries worldwide. As such, determining the business models and strategies that will ensure profitability, appropriate asset utilization and liquidity has been among the main challenges that businesses have faced on this path. Given that the effects of the pandemic are believed to be asymmetric across sectors, this study highlights the importance of undertaking a comparative analysis of the two sectors that are expected to face the most adverse effects of the crisis, and are anticipated to have different prospects in terms of coping with this turmoil. Aiming to assess the impact of COVID-19 on the key financial performance indicators that determine a business's capacity for financial sustainability and growth, this study applied asset and debt ratio analyses to the data of the top five companies in the world from both industries, in the time period from 2017 to 2022, intending to simultaneously examine the financial performance and sustainability of the e-commerce and hospitality industries in the last few challenging years.

The results of this study indicate that the assessed companies have shown the ability to successfully manage only some of their tangible and intangible assets, with some observed

variations across the firms in the analyzed period (2017–2022), regardless of the turmoil caused by the pandemic. According to the asset ratio analysis performed, one could observe that the companies effectively used inventory and accounts receivable, as these two types of assets have been identified as the key drivers of sales generation. On the contrary, the companies, especially hotels, have been struggling to effectively use fixed and total assets, as well as working capital, in the last few years. The assessed companies have developed distinctive policies in relation to their receivable collection. While some companies (Alphabet, Marriott and InterContinental, Hilton, Hyatt and Host Hotels & Resorts) have decided to enable better purchasing conditions through preserving or even extending the collection period and offering the accounts receivable privilege to their customers, being motivated to maintain their sales in the hard times of 2020, other companies (JD, Amazon and Meta) have decided to rush, urging payment collection with the intention of sustaining their liquidity. Regarding fixed asset management, the results suggest that their utilization for turnover generation was not much affected at the peak of the crises period in e-business, due to the fact that the lowest value of the fixed asset turnover ratio in 2020 among e-commerce companies was recorded only by Alphabet, while for the hotels, the situation was different, as, except for Hyatt, all other analyzed hotels recorded the lowest value of this ratio in 2020. The debt ratio analysis performed in this study has implied that the assessed companies in the hotel industry could not sustain the desired capital structure, and have reshaped it, increasing their reliance on debt in 2020 and 2021 to finance their assets. On the contrary, the selected e-commerce companies were found on average to rely less on debt to finance assets, whereby the highest values of their debt-to-equity ratio (excluding Alphabet and Meta) were identified in 2017 and 2018. As such, Marriott achieved the maximum value of debt-to-equity share among the hotels, while Alphabet reached the highest value of this ratio among e-businesses. When it comes to the capacity of the companies to meet their interest obligations, the results suggest that except for Amazon, JD, Marriott and Hilton, which did not generate sufficient operating incomes to cover these expenses, all other assessed companies could successfully manage it.

Summarizing the results of the study, it can be concluded that the assessed companies managed to successfully respond to the challenges that arose within the observed period, keeping the trend in their financial performance positive, mostly due to their attempts to preserve or upgrade their sales revenues, such as those in the e-commerce industry, thus maintaining their profitability levels over the entire examined period. However, in accordance with expectations, differences across the examined sectors and companies were observed, mostly in regard to the lower scale of the utilization of fixed assets to generate turnover and in terms of the increased share of debt used to finance assets in the hotel industry, which was among first and hardest hit by the pandemic. This could be explained by the fact that that the demand for hotel services is of greater sensitivity, given the risks of the recent pandemic, thus raising questions about revenue collection by low cost-to-revenue elasticity, due to the significant share of the fixed costs (such as wages and charges), which could not be significantly decreased in the short run, and by the fact that they were exposed to the additional costs of implementing health and safety protocols, such as increased cleaning and sanitation measures and the purchase of personal protective equipment for staff and guests. Altogether, this has given rise to reductions in cash reserves, thus forcing firms to take on more in debt to meet their financial obligations, as confirmed by the results of this study. In line with expectations, the e-commerce industry has proven to be more resilient to the new circumstances, proving to be able to better manage both assets and debt, mostly due to a sharp increase in their revenues being induced via a change in customer behavior. Nevertheless, the e-commerce sector was also faced with challenges, bearing in mind the enormous increase in the demand and shipping delays during the pandemic, which, together with the decreasing duration of the collection period, has limited their financial growth. An adequate response to the changed business circumstances and finance recuperation is a function of many factors in both sectors, and, as such, would require higher flexibility, better quality of customer service, satisfaction with customer experience,

the reestablishing of trust and feelings of safety, strong support by the government, and synergy between all relevant stakeholders.

This study is considered to make a dual contribution to the relevant scholarly literature. Firstly, based on the accounting data and financial ratio measures employed, the study provides the conditions for operationalizing the concept of financial sustainability in the observed companies, thus providing an important input in relation to the relevant managerial, investment and policy creation decisions and financial governance at the company and industry levels. Thus, the study expands the financial sustainability literature by developing a framework to capture and measure the objective of "sustainable value creation", but also fills the gap wherein no previous cross-sectional research has been conducted to simultaneously assess the impacts of COVID-19 on the financial safety of companies in the two sectors, which were given an asymmetric prognosis in relation to their financial performance and ability to secure financial sustainability in the observed period.

The limitations of the study are reflected in the fact that the research is focused on a few leading companies belonging to the assessed sectors, while there are no representatives of small and medium-sized firms in the sample, which may constrain the generalization. Also, some of the data required in the research were not available for all selected companies in the observed period. In addition, the study employed only selected financial ratios, focusing on the asset and debt aspects in drawing conclusions on the businesses' financial performance in the examined period. Moreover, it is worth mentioning that accounts payable management, within asset management, has not been considered due to the lack of accounts payable information for all the selected companies. Lastly, the study has not investigated the internal and external factors that may have impacted the adoption and usage of e-commerce mechanisms, barriers to the full exploitation of their benefits during the pandemic, as well as possible strategies to maximize the benefits and mitigate the burdens associated with the implementation of e-business models. Accordingly, the research could be further extended to fill these gaps, thus enhancing the quality of both the study's findings and its conclusions.

**Author Contributions:** Conceptualization, G.V. and J.S.; methodology, G.V. and J.S.; formal analysis, J.S.; resources, G.V.; data curation, J.S. and M.V.; writing—original draft preparation, G.V., J.S. and M.V.; writing—review and editing, G.V.; visualization, M.V. All authors have read and agreed to the published version of the manuscript.

**Funding:** This research received no external funding.

**Data Availability Statement:** Research data supporting reported results can be found at: https://ir.aboutamazon.com/annual-reports-proxies-and-shareholder-letters/default.aspx (accessed on 30 April 2023). https://abc.xyz/investor/ (accessed on 30 April 2023). https://ir.jd.com/annual-reports (accessed on 30 April 2023). https://www.alibabagroup.com/en-US/ir-financial-reports-financial-results (accessed on 30 April 2023). https://investor.fb.com/financials/default.aspx (accessed on 30 April 2023). https://marriott.gcs-web.com/annual (accessed on 30 April 2023). https://ir.hilton.com/financial-reporting/annual-reports (accessed on 30 April 2023). https://investors.hyatt.com/investor-relations/financial-reporting/annual-reports/default.aspx (accessed on 30 April 2023). https://ir.hosthotels.com/annual-reports (accessed on 30 April 2023). https://www.ihgplc.com/en/investors/annual-report (accessed on 30 April 2023).

**Conflicts of Interest:** The authors declare no conflict of interest.

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
