# Peer review of "Managing Financial Performance toward Achievements in Sustainability Prospects: Comparative Analysis of the e-Commerce and Hospitality Industries"

_jrfm, doi:10.3390/jrfm16090395_

Round 1

Reviewer 1 Report

28.07.2023.

Review report for: Managing financial performance towards sustainability prospects achievements: Comparative analysis of e-commerce and hospitality industry

Dear Authors,

I have read Your paper and I find it interesting. However, I have some suggestion for which I believe it could improve the paper.

(1) My main concern is suitability of this for special issue, therefore I would suggest to connect the paper more with sustainability. There are 3 pillars of sustainability: economic, social and environmental, and I would advise You to related Your research and results more with this economic sustainability. Title of the paper contains sustainability, so I would suggest to link the research with sustainability also in abstract, introduction and conclusion, but also to add some literature review related to economic sustainability.

(2) In abstract and introduction, please add that the analysis is conducted for the top five “worlds” companies, since until methodology we do not get information about companies included into sample

(3) In conclusion, please write clearly the answer on defined research questions: did or did not companies use their asset effectively, did or did no offer accounts receivable privileges, and has the leverage change during this period. I suggest to add average rations in Table 1 and Table 2 for two sectors, and based on that value conclusions above averages in the sectors could be made.

(4) check the sentence, I think something is missing: “According to the available data, the selected companies achieve sales per each dollar of inventory (inventory turnover ratio), accounts receivable (accounts receivable turnover ratio), fixed asset (fixed asset turnover 4ratio), working capital (sales to working capital), total assets (total assets turnover ratio).“ (raw 416)

(5) Please elaborate the contribution of this paper in conclusion

/

Author Response

Dear reviewer,

We are honored to have the opportunity to re-submit our manuscript named “Managing Financial Performance Towards Sustainability Prospects Achievements: Comparative Analysis of E-Commerce and Hospitality Industry” to this distinguished journal, acompassing the edits as per your comments.

Response:

(1) My main concern is suitability of this for special issue, therefore I would suggest to connect the paper more with sustainability. There are 3 pillars of sustainability: economic, social and environmental, and I would advise You to related Your research and results more with this economic sustainability. Title of the paper contains sustainability, so I would suggest to link the research with sustainability also in abstract, introduction and conclusion, but also to add some literature review related to economic sustainability.

With the outbreak of COVID-19, struggling to maintain the existence in the market has become one of the priorities of the business in all the industries worldwide. In this view, determining the business models and strategies for ensuring the financial sustainability measured through proper asset utilization and debt management has been among the main challenges that the businesses have been faced with on this path. Given that the effects of the pandemic are believed to be asymmetric across the sectors, and that according to the current literature the impact of by the pandemic caused economic crisis on particular sectors remains still a black box, this study is believed to be relevant as it employs a comparative analysis of the two sectors’ financial sustainability trends as these sectors are expected to have adverse effects from the crisis and are anticipated to have different prospects in coping with this turmoil. Nevertheless, a literature that simultaneously examines the impact of the recent pandemic on the financial performance and sustainability of the sectors which have been given the asymmetric prognosis, such as e-commerce and hospitality, is scarce. Aiming to fit this gap, this study intends to assess the effects of COVID-19 on the financial performance and sustainability of the top five companies in the world from each sector. This study focuses on the financial sustainability, being a part of the tree pillar sustainability concept described in Brundtland Commission Report (1987), aimed at ensuring the long term financial security of the observed companies and sustainable value creation (Unitaded Nations, 1987).

Accordingly, the concept of financial sustainability has been more profoundly linked to this research in the main sections of the report, according to the reviewer’s suggestions.

(2) In abstract and introduction, please add that the analysis is conducted for the top five “worlds” companies, since until methodology we do not get information about companies included into sample.

In line with the reviewer’s suggestion, it has been added the required content to the Abstract.

“The aim of the research is to simultaneously examine the financial performance and sustainability of the e-commerce and hospitality industries, employing the asset and debt ratio analysis conducted for the top five companies in the world from each sector from each sector, in the time period from 2017 to 2022.”

(3) In conclusion, please write clearly the answer on defined research questions: did or did not companies use their asset effectively, did or did no offer accounts receivable privileges, and has the leverage change during this period. I suggest to add average rations in Table 1 and Table 2 for two sectors, and based on that value conclusions above averages in the sectors could be made.

The answers to research questions have been added in conclusion and clarified, as requested. Namely, it is clarified what type of assets companies effectively use or not, which companies offer the accounts receivable privilege, as well as which companies managed to sustain a desirable capital structure or changed their financial leverage. As for the suggested average ratios in Table 1 and Table 2 for two sectors, due to the huge differences in ratios’ value among companies within the same sector and over the years, calculating average values would make it difficult to reach a conclusion.

(4) Check the sentence, I think something is missing: “According to the available data, the selected companies achieve sales per each dollar of inventory (inventory turnover ratio), accounts receivable (accounts receivable turnover ratio), fixed asset (fixed asset turnover 4ratio), working capital (sales to working capital), total assets (total assets turnover ratio).“ (raw 416)

The sentence is corrected as: According to the available data, the selected companies achieve sales greater than one per each dollar of inventory…”

(5) Please elaborate the contribution of this paper in conclusion

In line with the reviewer’s suggestion, it has been added the required content to the Conclusion.

“This study is considered to have a dual contribution to scholarly relevant literature. Firstly, based on the employed accounting data and financial ratio measures, the study provides the conditions for operationalizing the concept of financial sustainability in the observed companies, thus giving an important input for the relevant managerial, investment and policy creation decisions and financial governance at the company and industry level. Thus, the study expends the financial sustainability literature by developing a framework to capture and measure the objective of “sustainable value creation”, but also fits the gap of no previous cross-sectional research done to simultaneously assess the impact of COVID-19 on the financial safety of the companies from the two sectors, which are given the asymmetric prognosis on their financial performance in the observed period.”

Reviewer 2 Report

Thank you for allowing me to review this interesting paper. To get a better performance of the paper below are my proposals for improvements, and they are:

-        The research methodology needs to be defined scholarly, issues of validity and reliability are very questionable. The paper needs to be reworked driven by a clear purpose and its' importance.

-        The discussions often come from an assumptions about the reader, i.e they are aware of all possible information within this field. There are too many concepts that were added which are actually not appropriate for the paper.

-        The paper is not clear on the issues and problems this paper is hoping to solves. All the important variables need to be defined adequately. A relationship between these variables needs to be identified. How is this going to add value to knowledge in the future, for whom. All important variables are treated superficially, it is my opinion this is the case because too many variables are dealt without a real need or purpose.

-        Why are the financial performances of e-commerce and hospitality industry being examined or compared? Are there previous studies that show the need to compare the financial performance of e-commerce and the hospitality industry? State the reasons for comparing the financial performance of the e-commerce and hospitality industries, if any? If there is an assumption that e-commerce experienced the expansion of the hospitality industry in reverse, are there works that initiate the need to compare the financial performance of e-commerce and the hospitality industry?

-        The conducted analysis of the financial performance of the e-commerce and hospitality industry clearly raises the issue of separate observation of the e-commerce and hospitality industry sectors with an increase in the sample of observed companies. Due to the incomparability of these two sectors, the question of their organization and the validity of the research results is raised.

-        The hotel supply chain management has no direct connection to the subject of the research, nor the references cited in the paper in this regard.

Extensive editing of English language required

Author Response

Dear reviewer,

We are honored to have the opportunity to re-submit our manuscript named “Managing Financial Performance Towards Sustainability Prospects Achievements: Comparative Analysis of E-Commerce and Hospitality Industry” to this distinguished journal, acompassing the edits as per your comments.

Response:

(1) The research methodology needs to be defined scholarly, issues of validity and reliability are very questionable. The paper needs to be reworked driven by a clear purpose and its' importance.

The methodology has been defined based on the principles in financial management, often used as valid and reliable for financial statement analysis of companies, as well as their comparison to other companies, industry average, for following changes in companies’ financial position and performance. The importance of this methodology is reflected in advancing research in the field of sectors/industries that have been affected differently in the time of COVID-19 crisis, serving to the academic public for further investigation of crises repercussions to financial performance of companies/industries. In addition to the already existing elaboration of each ratio from assets and debt management analysis, as well as additional explanation of the importance of both conducted analysis, in line with the request, a clear methodology of ratios calculation has been added. 

(2) The discussions often come from an assumptions about the reader, i.e they are aware of all possible information within this field. There are too many concepts that were added which are actually not appropriate for the paper.

The research is based on the assumptions about the expected adverse, COVID-19 related, effects for the companies’ financial sustainability and performance in the two sectors which are anticipated to have different prospects in coping with this turmoil. The assumptions are based on the provided data about the two sectors’ performance and trends in the prior period and the differences in their sensitivity to the pandemic conditions. Considering the complexity of the relationships between the observed variables and the need for providing enough background information to more profoundly obtain insights in the processes and effects occurred in the two sectors, a voluminous literature has been reviewed and a number of concepts have been included to better shad the light over the possible causes and implications for the observed companies’ finance in the examined period.

(3) The paper is not clear on the issues and problems this paper is hoping to solves. All the important variables need to be defined adequately. A relationship between these variables needs to be identified. How is this going to add value to knowledge in the future, for whom. All important variables are treated superficially, it is my opinion this is the case because too many variables are dealt without a real need or purpose.

The repercussion of COVID-19 crises on these two sectors have been analyzed from the aspect of changes in their assets and capital structure in order to evaluate the companies’ ability to effectively use their assets, weather the changed business conditions affected their ability to offer the accounts receivable privilege, as well as how recent development affect their capital structure. Aiming to answer to these research questions, the assets and debt management analysis have been employed, with their most often used ratios (excluding those ratios which require not publicly available information). The selected ratios are related to each other belonging to the same type of financial statement analysis and provide a better understanding of the assets and debt management, where excluding some of them would form a gap in elaboration. The findings of this study are intended to provide an in-depth understanding of the two sectors’ financial vulnerability and their response the reshaped business environment and associated risks, which may be highly relevant to the observed companies’ management in an attempt to ensure positive financial results, to risk-averse investors in making investment decisions, but also to policy makers, aiming to mitigate the new risks and assess the future prospects of the companies, entire sectors and economies. By doing so, this paper aims to provide valuable insights into the economic consequences of the COVID-19 pandemic on the hotel and e-commerce industries and to identify potential strategies for recovery and resilience in the face of future crises.

(4) Why are the financial performances of e-commerce and hospitality industry being examined or compared? Are there previous studies that show the need to compare the financial performance of e-commerce and the hospitality industry? State the reasons for comparing the financial performance of the e-commerce and hospitality industries, if any? If there is an assumption that e-commerce experienced the expansion of the hospitality industry in reverse, are there works that initiate the need to compare the financial performance of e-commerce and the hospitality industry?

Given the findings of the current literature, according to which it is reasonable to anticipate that the severity of the damage across the industries has been driven by their vulnerability to pandemic conditions and capacity to implement business models and strategies to cope with the new business environment, one may conclude that the sectors such as e-commerce and hotel industry are assumed to have the reverse trends in their operations and finance during the recent pandemic. However, since such an expected scenario realization is driven by a number of factors that arise from the business, industry or a country level, which may cause an unexpected outcome, it is why the impact COVID-19 on particular sectors remains still a black box, despite expectations.

Nevertheless, a literature that simultaneously examines the impact of the recent pandemic on the financial performance and sustainability across the sectors is scarce. Accordingly, this study highlights the importance of making a comparative analysis of the two sectors which are expected to have adverse effects from the crisis and are anticipated to have different prospects in coping with this turmoil. Thus, this study intends to fit the gap of no previous cross-sectional research done to simultaneously assess the impact of COVID-19 on the financial safety of the companies from the two sectors, which are given the asymmetric prognosis on their financial performance in the observed period. Hence, the aim with comparative analysis of e-commerce, as a promising industry, and hospitality, as an industry deteriorated by the last crisis, is to explore the similarities and differences in managing financial performance from the aspect of asset and debt management within the function of business sustainability. Moreover, the findings of this study intend to provide an in-depth understanding of the two sectors’ financial vulnerability and their response the reshaped business environment and associated risks, which may be highly relevant to observed companies’ management in an attempt to ensure positive financial results, to risk-averse investors in making investment decisions, but also to policy makers, aiming to mitigate the new risks and assess the future prospects of the companies, entire sectors and economies. By doing so, this paper aims to provide valuable insights into the economic consequences of the COVID-19 pandemic on the hotel and e-commerce industry and to identify potential strategies for recovery and resilience in the face of future crises.

(5) The conducted analysis of the financial performance of the e-commerce and hospitality industry clearly raises the issue of separate observation of the e-commerce and hospitality industry sectors with an increase in the sample of observed companies. Due to the incomparability of these two sectors, the question of their organization and the validity of the research results is raised.

The research goal is to examine receint changes in assets and debt management of two industries that have experienced, based on the literature, a rise (e-commerce) and fall (hotels) in their service provision, thus benefiting (e-commerce) or not (hotels) from the changed economic circumstances. The research results confirmed differences in customer relationship since mostly hotels, trying to survive crises, have decided to enable better purchase conditions through preserving or even extending the collection period and offer the accounts receivable privilege to their customers, motivated to sustain their sales in the hard times of 2020, while e-commerce companies mostly have decided to rush with the payment collection. Further, the debt ratio analysis in this study has implied that the assessed companies in the hotel industry could not sustain the desired capital structure and have reshaped it, increasing the reliance on debt in 2020 and 2021 to finance their assets. On the contrary, the selected e-commerce companies were found on average to rely less on debt to finance assets.

(6) The hotel supply chain management has no direct connection to the subject of the research, nor the references cited in the paper in this regard.

The study provides an insight into the supply chain matters of the observed industries as it is noted that financial effects in the companies during the recent pandemic have been assumed to arise from both, the supply and the demand side negative shock. “Reinforcing each other, these shocks have brought many challenges in the functioning of the supply chains, international trade, investment flows and labor markets worldwide. Such an adversity, embodied in the resources’ deficiency, steep shrink in the volume of operations, alike collapse in the consumption and hence in the plunge of the sales turnovers and cash flows have implied serious repercussions on the corporate finance and financial sustainability.”